# Metabolome and Transcriptome Unveil the Correlated Metabolites and Transcripts with 2-acetyl-1-pyrroline in Fragrant Rice

**DOI:** 10.3390/ijms25158207

**Published:** 2024-07-27

**Authors:** Yu Zeng, Baoxuan Nong, Xiuzhong Xia, Zongqiong Zhang, Yuhao Wang, Yong Xu, Rui Feng, Hui Guo, Yuntao Liang, Can Chen, Shuhui Liang, Xianbin Jiang, Xinghai Yang, Danting Li

**Affiliations:** Key Laboratory of Rice Genetics and Breeding, Rice Research Institute, Guangxi Academy of Agricultural Science, Nanning 530007, China; zeng2019yu@163.com (Y.Z.); nbx1980@163.com (B.N.); xiaxiuzhong@163.com (X.X.); zhangzongqiong@gxaas.net (Z.Z.); wwm321a@163.com (Y.W.); xuyongen@hotmail.com (Y.X.); frlbl@163.com (R.F.); gh8207@163.com (H.G.); liangyt@sina.com (Y.L.); chencan129@126.com (C.C.); liangsh66@foxmail.com (S.L.); xp.jiang@163.com (X.J.)

**Keywords:** fragrant rice, metabolome, transcriptome, WGCNA analysis, 2AP

## Abstract

Fragrance is a valuable trait in rice varieties, with its aroma significantly influencing consumer preference. In this study, we conducted comprehensive metabolome and transcriptome analyses to elucidate the genetic and biochemical basis of fragrance in the Shangsixiangnuo (SSXN) variety, a fragrant indica rice cultivated in Guangxi, China. Through sensory evaluation and genetic analysis, we confirmed SSXN as strongly fragrant, with an 806 bp deletion in the *BADH2* gene associated with fragrance production. In the metabolome analysis, a total of 238, 233, 105 and 60 metabolic compounds exhibited significant changes at the seedling (S), reproductive (R), filling (F), and maturation (M) stages, respectively. We identified four compounds that exhibited significant changes in SSXN across all four development stages. Our analyses revealed a significant upregulation of 2-acetyl-1-pyrroline (2AP), the well-studied aromatic compound, in SSXN compared to the non-fragrant variety. Additionally, correlation analysis identified several metabolites strongly associated with 2AP, including ethanone, 1-(1H-pyrrol-2-yl)-, 1H-pyrrole, and pyrrole. Furthermore, Weighted Gene Co-expression Network Analysis (WGCNA) analysis highlighted the magenta and yellow modules as particularly enriched in aroma-related metabolites, providing insights into the complex aromatic compounds underlying the fragrance of rice. In the transcriptome analysis, a total of 5582, 5506, 4965, and 4599 differential expressed genes (DEGs) were identified across the four developmental stages, with a notable enrichment of the common pathway amino sugar and nucleotide sugar metabolism in all stages. In our correlation analysis between metabolome and transcriptome data, the top three connected metabolites, phenol-, 3-amino-, and 2AP, along with ethanone, 1-(1H-pyrrol-2-yl)-, exhibited strong associations with transcripts, highlighting their potential roles in fragrance biosynthesis. Additionally, the downregulated expression of the *P4H4* gene, encoding a procollagen-proline dioxygenase that specifically targets proline, in SSXN suggests its involvement in proline metabolism and potentially in aroma formation pathways. Overall, our study provides comprehensive insights into the genetic and biochemical mechanisms underlying fragrance production in rice, laying the foundation for further research aimed at enhancing fragrance quality in rice breeding programs.

## 1. Introduction

The flavor of rice (*Oryza sativa* L.) is closely tied to its aroma, making fragrant rice a top choice for consumers and accounting for around 20% of global rice trading due to its high popularity [1]. Rice breeders have focused extensively on understanding the molecular and biochemical factors behind this fragrance and how to enhance it [2]. However, the biosynthesis of aromatic compounds remains an exceedingly intricate process, and to date, researchers have identified hundreds of volatile compounds in fragrant rice [3].

Of them, 2-acetyl-1-pyrroline (2AP), the volatile compound responsible for imparting a ‘popcorn-like’ aroma to a wide range of cereal and food products, has been revealed as the pivotal and predominant component responsible for this aroma in rice [4,5,6]. Over the last two decades, scientific efforts have been dedicated to unraveling the biosynthetic mechanism of 2AP. Researchers have unveiled proline, glutamic acid, ornithine, and 1-pyrroline as crucial precursors for 2AP [5,7]. An enzyme, pyrroline-5-carboxylate synthetase (P5CS), was found as a key role in the biosynthesis of 2AP [8]. Additionally, genetic analysis reveals that the presence of rice fragrance is linked to the mutated *badh2* gene, which encodes putative betaine aldehyde dehydrogenase [9,10,11]. A further study has highlighted the significant role of the *BADH2* gene in balancing the 2AP and γ-aminobutyric aldehyde (AB-ald) content in fragrant rice [12]. The intact enzyme, betaine aldehyde dehydrogenase (BADH), hinders the conversion of AB-ald to 1-pyrroline by transforming AB-ald into γ-aminobutyric acid (GABA), while the existence of null *badh2* alleles led to the accumulation of AB-ald and a subsequent increase in 2AP biosynthesis [2]. The synthesis of 2AP is not only in a *BADH* dependent way. The study further found the involvement of P5CS and glyceraldehyde-3-phosphate dehydrogenase (*GAPDH*) in the accumulation of 2AP [13]. This suggests a potential reaction between ∆1-pyrolline-5-carboxylic acid and methylglyoxal, leading to 2AP formation. 

China is rich in fragrant glutinous rice germplasm resources and has an extensive historical legacy of rice cultivation. Over the long history of its domestication, fragrant glutinous rice has undergone adaptive diversification, leading to the emergence of ecotypes suitable for diverse regional agroecosystems [14,15]. Rice landraces, in contrast to modern rice cultivars, exhibit greater genetic complexity, a richer reservoir of genetic diversity, robust environmental adaptability, disease and pest resistance, high yield potential, and superior quality [16]. Guangxi province possesses an abundance of rice landraces [17,18]. In our previous study, out of the 179 local fragrant rice varieties we gathered from Guangxi province, a remarkable 97.77% were glutinous rice, totaling 175 varieties [19]. Local rice varieties possess a rich reservoir of genetic diversity, not only offering a valuable resource for plant breeders but also providing good material for research of molecular and biochemical factors behind some traits, like fragrance development [20,21]. 

Significant advancements and innovations have greatly improved gas chromatography (GC), particularly when it is integrated with mass spectrometry (MS), known as GC-MS. This integration has simplified the process of identifying and quantifying volatile aroma compounds (VACs) in complex sample matrices, substantially enhancing our understanding of the chemistry behind rice aroma [3]. Combining GC-MS and sensory evaluation, a study has found that 2AP and its structural homologs, along with various other consistently detected metabolites, play a crucial role in distinguishing fragrant Jasmine rice varieties, each contributing to distinct sensory properties [22]. Using multivariate analysis of GC-MS, fifteen volatile compounds were regarded as potential biomarkers for establishing the link between volatile compounds and fragrance [23]. Eight key marker compounds were identified, including pentanal, hexanal, 2-pentylfuran, 2,4-nonadienal, pyridine, 1-octen-3-ol, and (E)-2-octenal, as distinguish aromatic rice using GC-MS in Indonesia [24]. Hexanal, octanal, nonanal, (E)-2-octenal, decanal, 1-heptanol, and 1-octanol were identified as major aroma-active compounds in Jasmine rice [25]. GC-MS comes together as a potent instrument for complementing sensory evaluation, enabling us to explore the relationship between phenotypes and metabolites. This, in turn, helps us pinpoint the key compounds that define the aroma of fragrant rice. The identity of the compound and its role in 2AP synthesis have yet to be determined. Thus, there is still much to learn about the processes that regulate 2AP accumulation in fragrant rice grains.

The advent and utilization of advanced transcriptome sequencing, high-resolution metabolome, and information processing technologies have propelled the prevalence of systems biology (omics) research in investigating significant biological phenomena [26]. Notably, the integration of transcript and metabolite datasets via correlation and clustering analyses has facilitated the creation of interconnected networks that link genes and metabolites in rice [27,28,29]. However, there has been minimal research that combines transcriptome and volatile aroma metabolome analyses to explore the underlying mechanisms of rice fragrance metabolism.

In this study, our aim was to identify differentially expressed fragrant metabolites and genes in a local fragrant rice variety using metabolome and transcriptome analysis. Furthermore, we employed Weighted Gene Co-expression Network Analysis (WGCNA) analysis to investigate the relationships between metabolites and transcripts/gene expression. Finally, correlation analysis between phenotype-related metabolites and genes was conducted to gain insights into the network of fragrance formation. The objective of this research was to enhance our understanding of the pathways leading to aroma in rice.

## 2. Results

### 2.1. Rice Fragrance Confirmation through Sensory and Genetic Analysis

We conducted an aroma assessment using the rice of the non-fragrant HKG and fragrant SSXN (Figure 1A). Following the panel evaluation, the fragrance can be categorized into four groups: non-fragrant, slightly fragrant, moderately fragrant, and strongly fragrant. SSXN was classified as strongly fragrant, while the HKG was rated as non-fragrant. Further, we conducted sequencing of the *BADH2* gene in SSXN and HKG. Through BLAST analysis against the *BADH2* sequence of Nipponbare and its corresponding full-length mRNA, we found an 803 bp deletion in SSXN spanning from 1643 to 2448 bp downstream of the transcription start site (Figure 1B). This deletion encompasses 86 bp from exon 4, 697 bp from intron 4, and 23 bp from exon 5. The *BADH2* allele of SSXN was the badh2-E4-5.

### 2.2. Comparative Volatile Profiling in Fragrant and Non-Fragrant Rice Varieties

To identify the metabolites that related to the fragrance of SSXN, we performed metabolite profiling of volatile compounds including four development stages using GC-MS. Employing a non-fragrant variety, HKG, as a control, we conducted the pairwise comparison among all four stages. A total of 815 putative metabolites were identified among 48 samples (Appendix A). The biplot from principal components analysis (PCA) reveals that PC1 accounts for 24.5% of all the samples (Figure 2A). Notably, the samples exhibit a distinct separation based on sample type (grain versus leaf) along PC1, where the grain samples (F and M stages) form a cluster on the right side of the axis, and the leaf samples (S and R stages) were distributed on the left side. To pinpoint the metabolites demonstrating the most notable differences between fragrant and non-fragrant rice, we conducted an OPLS-DA analysis comparing two varieties across identical tissues (Figure 2B). The utilization of OPLS-DA sought to develop a robust rice classification system by capitalizing on differences in metabolites that distinctly separated fragrant (SSXN) and non-fragrant (HKG) varieties. Samples within each group, categorized based on different developmental stages, exhibited tightly knit clusters, demonstrating clear discrimination between the respective groups. The count of metabolites of significant changes (Variable Importance of Projection (VIP) ≥ 1, *p* value ≤ 0.05, and fold change ≥ 2) were 238 at the S stage, 233 at the R stage, 105 at the F stage, and 60 at the M stage (Appendix A). There were only four metabolites in common of the four-stage comparisons: 2-acetyl-1-pyrroline (2AP, Analyte268), (+)-epi-Bicyclosesquiphellandrene (Analyte536), ethanone, 1-(1H-pyrrol-2-yl)-(Analyte678), phenol, and 3-amino-(Analyte792) (Figure 2C, Table 1). With the sole exception of (+)-epi-Bicyclosesquiphellandrene, the remaining three metabolites were exclusively identified in SSXN. Moreover, these three metabolites demonstrated the highest expression levels during the M stage, suggesting a strong correlation between rice fragrance and these specific metabolites.

### 2.3. The Correlation Analysis of 2AP and Metabolome Co-Expression Analysis

To investigate the cooccurrence with the fragrant marker metabolite 2AP, we examined the correlation between 2AP and other metabolites (Appendix A). The metabolite exhibiting the highest positive correlation was Ethanone, 1-(1H-pyrrol-2-yl)-, with a correlation value of 0.99. Additionally, there were 12 other metabolites with correlation values exceeding 0.8, namely phenol, 3-amino-(0.98), sulfurous acid, hexadecyl 2-pentyl ester (0.95, Analyte345), unknown metabolite (0.90, Analyte569), naphthalene, 1,2,4a,5,8,8a-hexahydro-4,7-dimethyl-1-(1-methylethyl)-(0.89, Analyte566), benzofuran, 2,3-dihydro- (0.84, Analyte809), (3,4-Dimethoxyphenyl)-hydroxy-phenylacetic acid, 1-methylpiperidin-4-yl ester (0.84, Analyte543), 2-hydroxy-3-pentanone (0.83, Analyte286), 1-propanol, 2-methyl-(0.82, Analyte121), .alfa.-copaene (0.82, Analyte375), 1H-pyrrole, 2,5-dihydro-(0.8, Analyte379), pyrrole (0.8, Analyte387), and mepivacaine (0.8, Analyte795). These findings indicate a highly similar accumulation trend among these metabolites and 2AP.

To gain deeper insights into the dynamics of marker metabolite changes throughout the development of rice fragrance, we conducted WGCNA to explore the co-expression network of differentially expressed metabolites (DEMs). A total of 10 co-expression modules were identified (Figure 3A), characterized by their shared expression patterns (Appendix A, Appendix A). The correlation analysis of the sample and modules revealed a positive association between the maturation stage of SSXN and three modules, namely magenta, blue, and yellow (Figure 3B). Notably, the magenta and yellow modules exhibited positive correlations exclusively with SSXN-M, suggesting a potential association with the fragrance. The magenta module comprised 30 metabolites, including 13 aroma compounds, consisting of 9 esters, 2 ketones, 1 alcohol, and 1 aldehyde (Table 2). The yellow module contained 78 metabolites, including 31 aroma compounds, comprising 13 ketones, 9 aromatic compounds, 4 esters, 3 alcohols, 1 aldehyde, and 1 hydrocarbon (Table 3). Of interest is the observation that the three significantly upregulated metabolites (2-acetyl-1-pyrroline, ethanone, 1-(1H-pyrrol-2-yl)-, and phenol, 3-amino-) identified in the four-stage comparisons were intricately clustered within the yellow module. These findings suggest a correlation among the metabolites within these two modules, indicating their potential contribution to the distinctive fragrance of SSXN. To mine the relationship of these metabolites, we performed a co-expression network analysis of these two modules. In the magenta module, there were nine metabolites exhibited a co-expressed relationship (Figure 3C). The hub fragrant metabolites were the two ketones, which were 2-octen-4-one, 2-methoxy- (Analyte233) and 4-methylheptane-3,5-dione (Analyte130). In the yellow module, 28 fragrant metabolites showed a co-expressed relationship (Figure 3D). The top 10 connected metabolites included 3 alcohols (1-butanol, 3-methyl- (Analyte188), 1-butanol, 2-methyl- (Analyte187), 1-propanol, 2-methyl-(Analyte121)), 5 ketones (2-heptanone (Analyte166), 3-pentanone, 2,4-dimethyl- (Analyte92), 2,3-butanedione (Analyte57), 2-hydroxy-3-pentanone (Analyte286), 2,3-hexanedione (Analyte137)), and 1 aromatic compound (naphthalene, 1,2,4a,5,8,8a-hexahydro-4,7-dimethyl-1-(1-methylethyl)-, [1S-(1,4a,8a)]- (Analyte566)).

### 2.4. Comparative Transcriptome Profiling in Fragrant and Non-Fragrant Rice Varieties

To unravel the transcriptional landscape underlying fragrance formation in SSXN, we conducted a transcriptome analysis using the same samples employed in the metabolome study. A snapshot of the transcriptome was obtained by applying PCA. In contrast to the metabolome, the leaf samples clustered closely on the left side, while the grain samples were segregated on the right side (Figure 4A). Notably, the samples displayed a clear separation based on the variety (HKG versus SSXN). These findings underscore the distinct gene regulation patterns between fragrant and non-fragrant varieties. Among the 24 samples, we identified 24,399 expressed genes (Appendix A). The comparisons between varieties in four stages were performed. There were 5582, 5506, 4965, and 4599 differential expressed genes (DEGs) (q value ≤ 0.05, and fold change ≥ 2) in four stages, respectively (Figure 4B).

We performed GO and KEGG analyses for the DEGs, offering valuable insights into the annotations of genes involved in biological processes, molecular functions, and cellular components (GO analysis), as well as those associated with cellular processing, environmental information processing, genetic information processing, metabolism, and organismal systems (KEGG analysis). The top 10 enriched pathways of the DEGs from the four-stage comparisons are illustrated in Figure 5. In the seedling stages (Figure 5A), phenylpropanoid biosynthesis, biosynthesis of secondary metabolites, diterpenoid biosynthesis, flavonoid biosynthesis, phenylalanine biosynthesis, amino sugar and nucleotide sugar metabolism, flavone and flavonoid biosynthesis, mismatch repair, DNA replication, and cutin, suberine, and wax biosynthesis emerged as the most enriched pathways. In the reproductive stage (Figure 5B), diterpenoid biosynthesis, phenylalanine biosynthesis, Biosynthesis of secondary metabolites, phenylpropanoid biosynthesis, amino sugar and nucleotide sugar metabolism, glucosinolate biosynthesis, tryptophan metabolism, cyanoamino acid metabolism, flavonoid biosynthesis, and MAPK signaling pathway-plant were identified as the top 10 enriched pathways. For the filling stage (Figure 5C), the top 10 pathways enriched in the DEGs included tryptophan metabolism, DNA replication, amino sugar and nucleotide sugar metabolism, glucosinolate biosynthesis, alpha-linolenic acid metabolism, flavonoid biosynthesis, mismatch repair, betalain biosynthesis, benzoxazinoid biosynthesis, and glutathione metabolism. Finally, in the maturation stage (Figure 5D), the top 10 significantly enriched pathways comprised mismatch repair, glutathione metabolism, fatty acid metabolism, amino sugar and nucleotide sugar metabolism, biosynthesis of unsaturated fatty acids, carotenoid biosynthesis, homologous recombination, DNA replication, flavone and flavonoid biosynthesis, and arachidonic acid metabolism. Notably, only one pathway, amino sugar and nucleotide sugar metabolism, was enriched across all four stages.

### 2.5. Comparative Transcriptome Profiling in Fragrant and Non-Fragrant Rice Varieties

To explore deeper into the transcriptome regulation throughout the development of rice fragrance, we conducted a WGCNA analysis of all the expressed transcripts. Using the power value of 8, 18 co-expression modules were identified, each distinguished by their unique expression profiles (see Appendix A and Appendix A). The gene number ranged from 1 to 4700 across the 18 modules (Figure 6A). Upon careful examination of the sample modules, two significant correlations stood out: first, the steel blue module displayed a positive correlation with SSXN and a negative correlation with HKG (Figure 6B). Notably, this positive correlation strengthened from the seedling to the maturation stages. Second, considering that the production of 2AP is regulated by the dysfunction of the *BADH2* gene, we found the pale turquoise module intriguing. This module exhibited a negative correlation with SSXN and a positive correlation with HKG (Figure 6B); it is worth mentioning that the *BADH2* gene (LOC_Os08g32870) was contained in this module. We performed functional enrichment of the genes of these two modules. There were three significant enriched pathways in the steel blue module, which were ribosome, proteasome, and splicesome (Figure 6C). Four pathways, glycine, serine and threonine metabolism, panthothenate and CoA biosynthesis, glucosinolate biosynthesis, and biosynthesis of amino acids, were significantly enriched in the pale turquoise module (Figure 6D).

### 2.6. The Correlation between Metabolome and Transcriptome

To investigate the association between DEMs and DEGs underlying rice fragrance, we focused on metabolites from the magenta and yellow modules, and transcripts from the steel blue and pale turquoise modules for correlation analysis. This analysis revealed 40 metabolites and 432 transcripts exhibiting significant positive or negative correlations (Appendix A). In our analysis, we identified 466 significant positive correlations (correlation value > 0.95) between 30 metabolites and 133 transcripts (Figure 7A). phenol-, 3-amino-, 2AP, and ethanone, 1-(1H-pyrrol-2-yl)- emerged as the top three connected metabolites, with 66, 57, and 53 links, respectively. Three genes exhibited the highest positive correlations with other metabolites, each having 10 links. Among these, two genes were unannotated, while the third was identified as *PUB23*, an E3-ubiquitin-protein ligase. Additionally, we observed 459 significant negative correlations (correlation value < −0.6) involving 35 metabolites and 84 transcripts (Figure 7B). An exception was the *BADH2* gene depicted in Figure 7B, where despite its correlation value higher than −0.6. The top three most connected metabolites were oxime-, methoxy-phenyl-, phenol-, 3-amino-, and hexane, 3,3,4,4-tetrafluoro- with 72, 53, and 47 links, respectively. Likewise, the top three most connected transcripts were *P4H4*, *BIN4*, and *TAD2*, with 23, 17, and 16 links, respectively. Notably, the *P4H4* gene, which encodes a procollagen-proline dioxygenase that specifically targets proline, demonstrated downregulated expression in SSXN.

## 3. Discussion

Xiangnuo (XN), translating to “fragrant glutinous rice” in Chinese, represents a distinctive indica cultivar primarily cultivated in the ethnic minority regions of Guangxi, China (Figure 1A). The confirmation of fragrance in the SSXN variety through sensory evaluation, coupled with genetic analysis, would provide valuable insights into the genetic basis of aroma production in this variety. Our study identified SSXN as strongly fragrant, contrasting with the non-fragrant classification of the HKG variety. Genetic sequencing revealed a significant 806 bp deletion in the *BADH2* gene of SSXN, spanning exon 4, intron 4, and exon 5 (Figure 1B). This finding aligns with previous research, which also identified the same deletion in the *BADH2* gene in the five Xiangnuo cultivars of Guangxi [30]. The fragrance characteristic of rice is predominantly associated with the loss-of-function mutation in the BADH2 gene, resulting in the accumulation of 2-acetyl-1-pyrroline (2AP) [12]. 2AP stands as the extensively investigated compound responsible for imparting the characteristic fragrance to rice grains [6]. Notably, in the case of SSXN, comparative analysis with the non-fragrant HKG variety revealed a significant upregulation in 2AP content (Table 1). Conversely, the expression levels of transcripts associated with the BADH2 gene exhibited pronounced downregulation (Appendix A). The findings suggest that SSXN adheres to the well-studied phenomenon of fragrance production in rice, wherein mutations in the BADH2 gene disrupt the conversion of GABA, consequently leading to the accumulation of the aromatic compound 2AP.

2AP stands as a pivotal marker distinguishing aromatic rice from non-aromatic counterparts. Numerous prior investigations have delved into the mechanisms underlying the formation of 2AP [7,8,31]. In addition to 2AP, the aromatic profile of rice grains is significantly influenced by the composition and proportion of volatile compounds [32,33]. The exploration of metabolites correlated with 2AP emerges as a promising way to advance our understanding of fragrance biosynthesis across diverse rice varieties [22,23,34]. Our analysis identified some significant correlations between 2AP and various metabolites in SSXN (Appendix A). Ethanone, 1-(1H-pyrrol-2-yl)- displayed the highest positive correlation coefficient with 2AP, implying a potential role in aroma development. Additionally, we identified 13 metabolites with correlation coefficients surpassing 0.8, indicating a closely aligned accumulation pattern with 2AP. This collective evidence suggests a concerted metabolic network governing aroma biosynthesis in rice, where these metabolites may play crucial roles alongside 2AP. It is interesting to note that ethanone, 1-(1H-pyrrol-2-yl)- and pyrrole have been reported to correlate with 2AP in other studies. A metabolome of a set of elite aromatic rice varieties shows four amine heterocycles—6-methyl, 5-oxo-2,3,4,5-tetrahydropyridine (6M5OTP), 2-acetylpyrrole, pyrrole, and 1-pyrroline—that correlate strongly with 2AP [34]. Considering that ethanone, 1-(1H-pyrrol-2-yl)- is also known as 2-acetylpyrrole, this finding partly aligns with our results. Additionally, differential metabolite analysis comparing three most popular aromatic rice varieties with a non-aromatic variety has identified compounds such as acetoin, 2-methyloctylbenzene, bicyclo[4.4.0]dec,1-ene-2-isopropyl-5-methyl-9-methylene, and 2-methylfuran, further highlighting the complex interplay between metabolites in determining rice aroma profiles [35]. The presence of 2-acetylpyrrole, 1H-Pyrrole, 2,5-dihydro-, and pyrrole alongside 2AP as heterocyclic and aromatic compounds suggests a close biochemical relationship. Pyrroles always contribute to the characteristic aroma and flavor of various cooked foods, including meats, bread, coffee, and chocolate [36]. In our research, the existence of 2-acetylpyrrole, 1H-Pyrrole, 2,5-dihydro-, and pyrrole suggest their potential contribution to the overall aroma profile. The reported oxidation of 2AP to 2-acetylpyrrole at room temperature provides an explanation for the observed correlation between these two compounds [37]. The involvement of pyrrole in the biosynthesis of 2AP further supports the strong correlation observed in our analysis [34]. Our findings underscore the relation of aroma biosynthesis pathways in rice grains, where multiple compounds, including 2AP and related heterocyclic aromatic compounds, work in concert to contribute to the overall aroma profile.

Ester, ketone, alcohol, aldehyde, and aromatic compounds have been reported to contribute to the fragrance of rice [38,39]. In our WGCNA analysis, the magenta and yellow modules exhibited a positive correlation, especially with the maturation stage of SSXN (Figure 3). The magenta module, consisting of 30 metabolites, emerged as a particularly promising candidate for contributing to the fragrance of SSXN. Within this module, 13 aroma compounds were identified, comprising 9 esters, 2 ketones, 1 alcohol, and 1 aldehyde (Table 2). Similarly, the yellow module, comprising 78 metabolites, further enriched our understanding of the compounds associated with fragrance in SSXN. Among the diverse array of metabolites in this module, 31 aroma compounds were identified, including 13 ketones, 9 aromatic compounds, 4 esters, 3 alcohols, 1 aldehyde, and 1 hydrocarbon (Table 3). These metabolites impart a wide array of flavors, including fruity, floral, sweet, creamy, buttery, cheesy, nutty, popcorn-like, woody, chocolatey, caramel notes, and so on. The presence of such a diverse range of aroma compounds within the magenta and yellow modules underscores the complex nature of fragrance biosynthesis in SSXN.

Multiple studies have used metabolome to mine the metabolites associated with rice fragrance [22,35,39]. In this study, we also employed transcriptome analyses alongside metabolome to explore the novel insights of gene expression underlying fragrance development in fragrant rice. The pathway, amino sugar, and nucleotide sugar metabolism were enriched using the DEGs across all four stage comparisons (Figure 5). Nucleotide sugar metabolism is involved in the conversion of sucrose to starch and the accumulation of carbohydrate accumulation in rice [40,41]. Starch would interact with aromatic compounds. Interestingly, starch has been implicated in forming complexes with aroma compounds, such as amylose, a component of starch that can encapsulate aroma molecules, influencing their release and perception [38]. The amino sugar or nucleotide sugar may indirectly contribute to aroma retention and perception in fragrant rice through its role in starch biosynthesis and carbohydrate accumulation. Further studies are required to provide more evidence regarding the specific functions of amino sugar and nucleotide sugar metabolism in rice fragrance formation. Of particular note is the enrichment of the glutathione metabolism pathway in the DEGs observed during the maturation stage. Glutathione metabolism is known to play a critical role in conferring resistance to salt stress [42]. Interestingly, fragrant rice varieties have been reported to exhibit higher salt tolerance, attributed to the loss of *BADH*2 function [43,44]. The enrichment of glutathione metabolism in the maturation stage DEGs suggests a potential link between this metabolic pathway and the dysfunction of the *BADH2* gene in SSXN.

We also conducted the correlation analysis to elucidate the relationships between metabolites and gene expression patterns underlying rice fragrance in the SSXN variety. To emphasize the fragrance phenotype, we specifically selected the two positive modules from the metabolome data and one positive and one negative module from the transcriptome data (Figure 7). Among the top six connected metabolites, 2AP, ethanone, 1-(1H-pyrrol-2-yl)-, and oxime-, methoxy-phenyl- were characterized by popcorn flavor [34,39]. These results suggest that the correlated transcripts may play a significant role in the formation of fragrance in rice. Among these transcripts, the most highly connected gene was *P4H4*, encoding a prolyl hydroxylase that specifically utilizes proline to generate hydroxyproline [45]. Previous studies have demonstrated that proline can induce the accumulation of 2AP in rice [46,47]. Therefore, we hypothesize that the downregulation of the *P4H4* gene could inhibit the formation of hydroxyproline, consequently influencing the proline content. This, in turn, may affect the synthesis and accumulation of 2AP and other fragrant metabolites.

## 4. Materials and Methods

### 4.1. Material and Planting

We obtained the materials from the germplasm base of Guangxi Zhuang Autonomous Region Academy of Agricultural Sciences. One local fragrant variety, Shangsixiangnuo (SSXN), and one non-fragrant variety, Huakegu (HKG), were planted in Nanning, Guangxi, China, in July 2020. The Shangsixiangnuo variety is glutinous rice with a strong fragrance [19], which was grown widely among the Zhuang and Yao minority nations in Southern China. The samples were harvested at four periods: the seedling stage (S), the reproductive stage (R), the grain-filling stage (F), and the maturation stage (M), according to the rice development. The leaves were collected in the seedling and reproductive stages, and the grains were collected in the grain-filling and mature stages. The samples were collected enough for all experiments, including three biological replicates.

### 4.2. Fragrance Determination

To confirm the fragrant traits of the two varieties, a fragrance determination was performed first. The assessment of fragrance in rice leaves followed the KOH method [48]. Approximately 2 g of fresh leaves harvested during the tillering stage were sliced into pieces and placed in a conical flask. Each sample was then subjected to incubation with 10 mL of a 1.7% KOH solution at room temperature for 10 min. Subsequently, the samples were evaluated for fragrance by three individuals immediately after the flask lids were removed.

### 4.3. Sequence Analysis of Fragrant Gene

The genetic analysis of the fragrant control gene, *Badh2*, was performed to further verify the fragrant ability of the SSXN variety. The leaf of rice was ground into powder with liquid nitrogen. Genomic DNA extraction and purification of the respective strain were performed using the Biospin Plant Omni Genomic DNA Extraction Kit (Bioer, Hangzhou, China) following the manufacturer’s instructions. The PCR primers and amplification protocol to sequence the *Badh2*/*badh2* gene followed former research [49]. The PCR reaction mixture consisted of 20 μL, including 1 μL of each forward and reverse primers, 1 μL of genomic DNA, 10 μL of 2X EsTaq master mix (CW Century, Beijing, China), and 7 μL of ddH_2_O. The PCR reaction conditions involved three steps: initial denaturation at 94 °C for 2 min, followed by 30 amplification cycles consisting of denaturation at 94 °C for 30 s, annealing at 55 °C for 30 s, extension at 72 °C for 1 min, and a final extension at 72 °C for 5 min. The PCR products with corresponding product sizes were validated by agarose gel electrophoresis. DNA fragments were purified using a TIANgel Midi purification kit (Tiangen, Beijing, China) and then cloned into a pUC-T vector (Takara, Dalian, China) for further Sanger sequencing (Tsingke, Guangzhou, China). The sequences were assembled using ContigExpress software v. 11.3 (Invitrogen, Waltham, MA, USA). The assembled sequence of SSXN was aligned with the sequence of non-fragrant variety Nipponbare using AlignX software v. 11.3 (Invitrogen, Waltham, MA, USA). The full gene sequence of the rice *Badh2* gene (LOC_Os08g32870) was also obtained from the Rice Genome Annotation Project website.

### 4.4. Gas Chromatography–Time-of-Flight Mass Spectrometry Analysis and Data Annotation 

The samples of the four developmental stages were used. Each sample included six biological replicates. Approximately 50 mg of each sample was extracted with 450 μL of an extraction solution (consisting of a 3:1 ratio of methanol to water). Adonitol was spiked in as an internal standard. The volatile compounds were analyzed using a Pegasus High Throughput TOF-MS (GC-TOF-MS) (LECO, St. Joseph, MI, USA). The GC-TOF-MS analysis was conducted using an Agilent 7890 gas chromatograph system that was coupled with a Pegasus HT time-of-flight mass spectrometer. A DB-5MS capillary column, coated with 5% diphenyl cross-linked with 95% dimethylpolysiloxane (30 m × 250 μm inner diameter, 0.25 μm film thickness; J&W Scientific, Folsom, CA, USA), was employed in the system. An aliquot of 1 μL of the analyte was introduced in spitless mode. Helium was employed as the carrier gas, with a front inlet purge flow of 3 mL min^−1^ and a gas flow rate through the column of 1 mL min^−1^. The initial temperature was maintained at 50 °C for 1 min, followed by an increase to 310 °C at a rate of 10 °C min^−1^, and then held at 310 °C for 8 min. The temperatures of the injection, transfer line, and ion source were set at 280, 280, and 250 °C, respectively. The energy used was −70 eV in electron impact mode. Mass spectrometry data were acquired in full-scan mode within the *m*/*z* range of 50–500 at a rate of 12.5 spectra per second after a solvent delay of 6.17 min.

The Chroma TOF 4.3X software from LECO Corporation and the LECO-Fiehn Rtx5 database were employed for extracting raw peaks, filtering data baselines, calibrating the baseline, aligning peaks, conducting deconvolution analysis, identifying peaks, and integrating peak areas. The metabolite identification process considered both mass spectrum matching and retention index matching. Peaks that were detected in less than 50% of the quality control (QC) samples or had an RSD greater than 30% in QC samples were eliminated.

### 4.5. Ribosomal Nuclear Acids (RNA) Extraction, Library Construction and Sequencing

The corresponding samples used in the metabolome analysis were also used for the transcriptome analysis. Each sample included three biological replicates. Total RNA was extracted using a Trizol reagent kit (Invitrogen, Carlsbad, CA, USA) according to the manufacturer’s protocol. RNA quality was assessed on an Agilent 2100 Bioanalyzer (Agilent Technologies, Palo Alto, CA, USA) and checked using RNase free agarose gel electrophoresis. After total RNA was extracted, eukaryotic mRNA was enriched by Oligo(dT) beads. Then, the enriched mRNA was fragmented into short fragments using fragmentation buffer and reverse transcripted into cDNA using ProtoScript II First Strand cDNA Synthesis Kit (NEB, Ipswich, MA, USA) with random primers, following the manufacturer’s protocol. Second-strand cDNA was synthesized by DNA polymerase I, RNase H, dNTP, and buffer. Then, the cDNA fragments were purified with a QiaQuick PCR extraction kit (Qiagen, Venlo, The Netherlands), end-repaired, poly(A) added, and ligated to Illumina sequencing adapters. The ligation products were size selected by agarose gel electrophoresis, PCR amplified, and sequenced using Illumina NovaSeq 6000 by Gene Denovo Biotechnology Co. (Guangzhou, China).

### 4.6. Quantitative Real-Time Polymerase Chain Raction (PCR) Validation

The transcriptome data underwent validation through quantitative real-time PCR (qRT-PCR) analysis, including 10 genes with upregulated or downregulated. The expression levels of these genes were normalized against the reference gene of cyclophilins (CYP). Primers were designed by Primer Premier 5.0 software, with details provided in Appendix A. For cDNA synthesis, HiScript II Q RT SuperMix (Vazyme, Nanjing, China) was employed following the manufacturer’s protocol. Each qRT-PCR reaction mixture (20 μL) contained 1 μL of cDNA, 10 μL of ChamQ Universal SYBR qPCR Master Mix (Vazyme, Nanjing, China), 0.4 μL of forward primer, 0.4 μL of reverse primer, and 8.2 μL of H_2_O. The qRT-PCR protocol included a pre-denaturing step (95 °C for 3 min), followed by amplification steps (95 °C for 5 s, 58 °C for 15 s, 72 °C for 20 s, for 40 cycles), and a melting curve analysis (continuous fluorescence capture from 60 °C to 95 °C). The qRT-PCR was conducted using an ABI step one plus a Real-Time Thermal Cycler (Thermo Fisher, Waltham, MA, USA). Relative gene expression was calculated utilizing the 2^−ΔΔCT^ method [50]. To validate the RNA-seq data, qRT-PCR analysis was performed on 10 randomly chosen transcripts (Appendix A).

### 4.7. Data Analysis

The sequencing reads underwent initial filtration using fastp (version 0.18.0) [51]. Bowtie2 (version 2.2.8) was utilized for aligning the reads to the ribosomal RNA (rRNA) database [52]. Clean reads remaining after filtration and rRNA sequence removal were employed for assembly and gene abundance calculation. HISAT2 (version 2.4) was employed to build an index of the reference genome of *Oryza sativa* ssp. *japonica* cv. Nipponbare, and paired-end clean reads were subsequently mapped to this reference genome with the parameter “-rna-strandness RF” and default settings [53]. The mapped reads from each sample were assembled using StringTie v1.3.1 via a reference-based approach [54,55]. Expression abundance and variations for each transcription region were quantified using StringTie software, calculating FPKM (fragments per kilobase of transcript per million mapped reads) values. Differential expression analysis of RNAs was conducted using DESeq2 software v. 1.45.1 between different groups [56], considering genes/transcripts with a false discovery rate (FDR) below 0.05 and an absolute fold change ≥ 2 as differentially expressed. Correlation analysis was carried out using R, while principal component analysis (PCA) was performed with the R package models (http://www.r-project.org/ accessed on 2 February 2024). For functional annotation, BLASTX v2.2.26 searches were performed against several public databases, including COG, KEGG, KOG, Pfam, Swiss-Prot, and NR, with a cutoff E-value of ≤1 × 10^−5^. Functional information from the hit with the highest total score was then assigned to the corresponding query.

## 5. Conclusions

Here, we studied the comparative metabolome and transcriptome between a fragrant variety and a non-fragrant variety of rice in four development stages. Our study provides comprehensive insights into the genetic and metabolic basis of fragrance in a variety of rice. Through a combination of metabolome and transcriptome analyses, coupled with WGCNA and correlation analyses, we elucidated key pathways and molecular components associated with fragrance metabolite production. Our findings not only underscore the pivotal role of mutations in the *BADH2* gene, which result in the accumulation of 2AP, but also shed light on the involvement of other metabolites and transcripts contributing to the fragrance of rice grains. Additionally, transcriptome analysis revealed differential expression patterns of genes related to fragrance metabolites, further enhancing our understanding of the regulatory mechanisms underlying aroma development in rice.

## Figures and Tables

**Figure 1 ijms-25-08207-f001:**
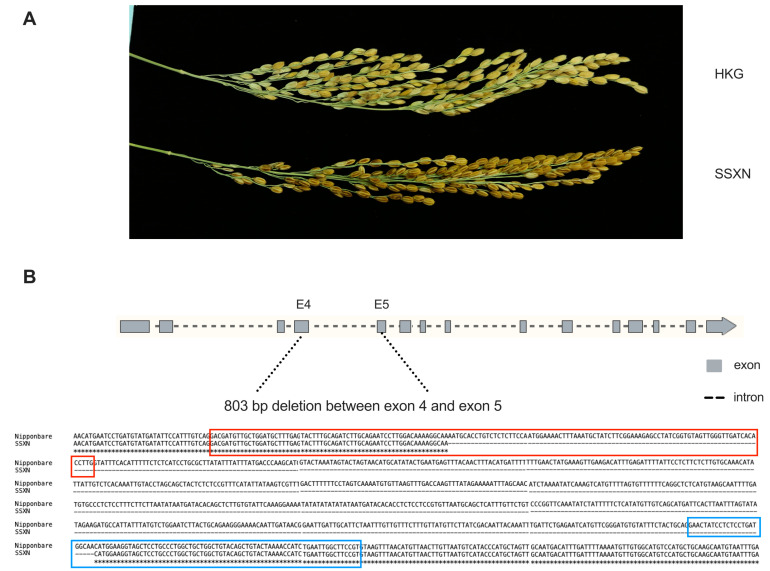
The grains and structure of *BADH2* gene in SSXN. (**A**) The grains of HKG (**upper**) and SSXN (**lower**). (**B**) The deletions between exons 4 and 5 were shown by alignment with the corresponding sequence of Nipponbare. The red frame indicates the exon 4, while the blue frame indicates the exon 5.

**Figure 2 ijms-25-08207-f002:**
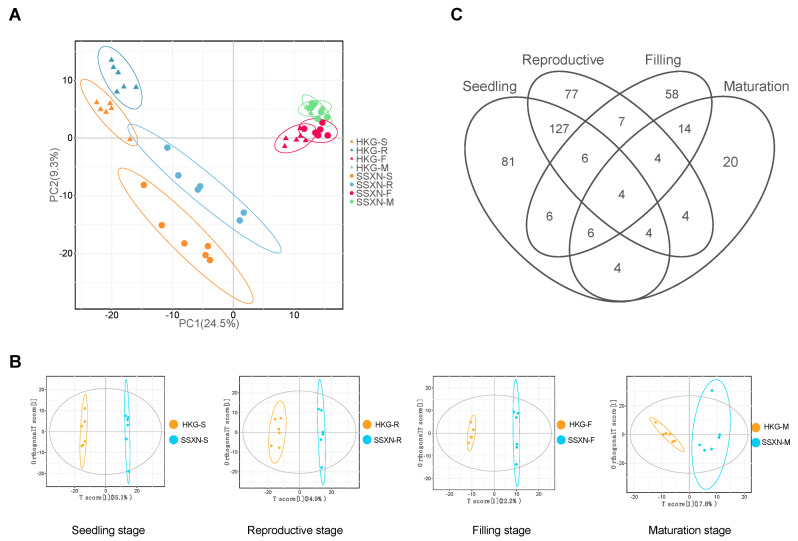
Multivariate analysis of SSXN and HKG varieties. (**A**) PCA analysis of metabolites from the two varieties across four stages. HKG is represented by triangles, while SSXN is represented by circles. Each stage is depicted in a different color. (**B**) OPLS-DA score plot comparing the four stages between SSXN and HKG. HKG is shown in orange, and SSXN is shown in blue. (**C**) Venn plot illustrating the differentially expressed metabolites across the four stages.

**Figure 3 ijms-25-08207-f003:**
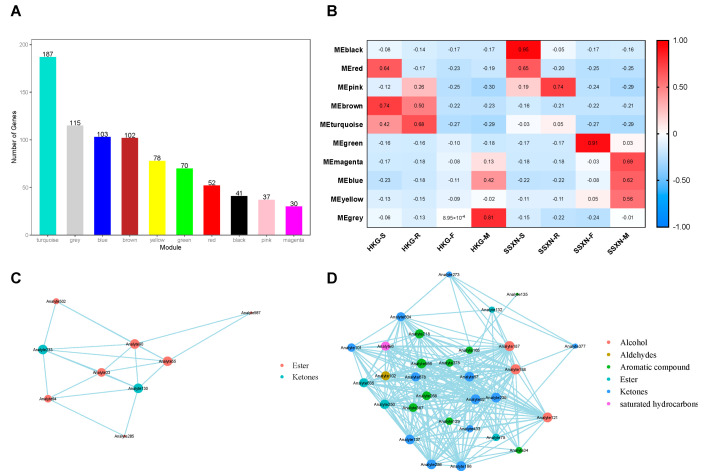
WGCNA analysis of metabolome, correlation relationship between modules and varieties, and metabolites co-expressed network. (**A**) Distribution of metabolites across 10 modules clustered by WGCNA analysis. (**B**) Correlation analysis of samples and modules. Values in the box represent the correlation between modules and varieties, with positive correlations depicted in red and negative correlations in blue. (**C**) Co-expression network of metabolites in the magenta module. Each circle represents a metabolite, with different types of metabolites depicted in different colors. (**D**) Co-expression network of metabolites in the yellow module.

**Figure 4 ijms-25-08207-f004:**
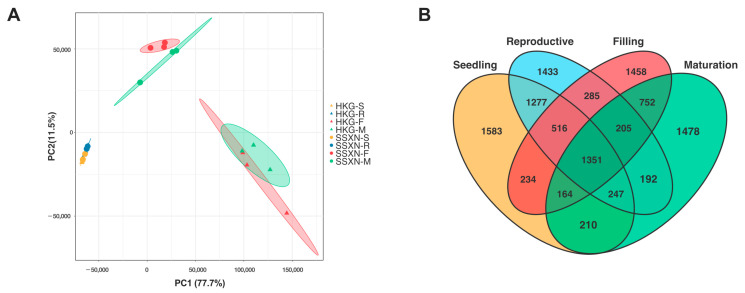
The differential analysis of transcriptome. (**A**) PCA analysis of transcripts from the two varieties across four stages. HKG is represented by triangles, while SSXN is represented by circles. Each stage is depicted in a different color. (**B**) Venn plot illustrating the differentially expressed transcripts across the four stages.

**Figure 5 ijms-25-08207-f005:**
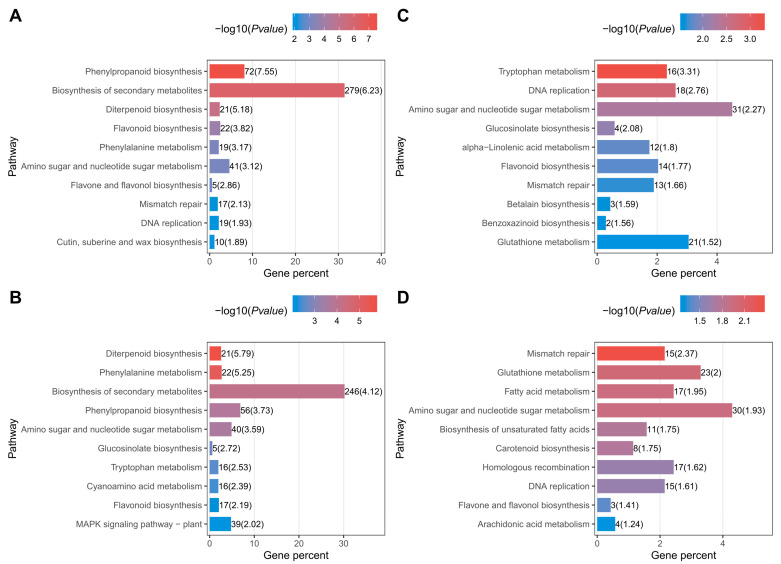
KEGG enrichment analysis of DEGs in the four stages. (**A**) The KEGG pathways enriched using the DEGs between the SSXN and HKG in seedling stage. (**B**) The KEGG pathways enriched using the DEGs between the SSXN and HKG in reproductive stage. (**C**) The KEGG pathways enriched using the DEGs between the SSXN and HKG in grain filling stage. (**D**) The KEGG pathways enriched using the DEGs between the SSXN and HKG in maturation stage.

**Figure 6 ijms-25-08207-f006:**
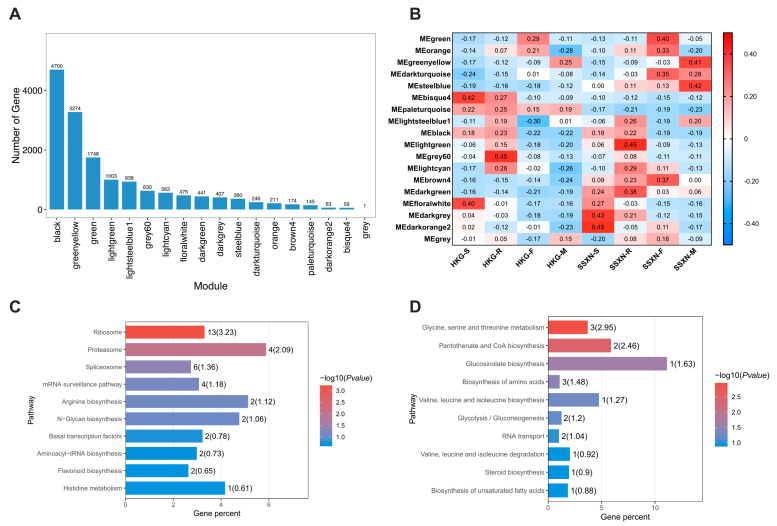
WGCNA analysis of transcriptome, correlation relationship between modules and varieties, and KEGG enrichment analysis of two modules. (**A**) Distribution of transcripts across 10 modules clustered by WGCNA analysis. (**B**) Correlation analysis of samples and modules. Values in the box represent the correlation between modules and varieties, with positive correlations depicted in red and negative correlations in blue. (**C**) The KEGG pathways enriched using the transcripts in the steel blue module. (**D**) The KEGG pathways enriched using the transcripts in the pale turquoise module.

**Figure 7 ijms-25-08207-f007:**
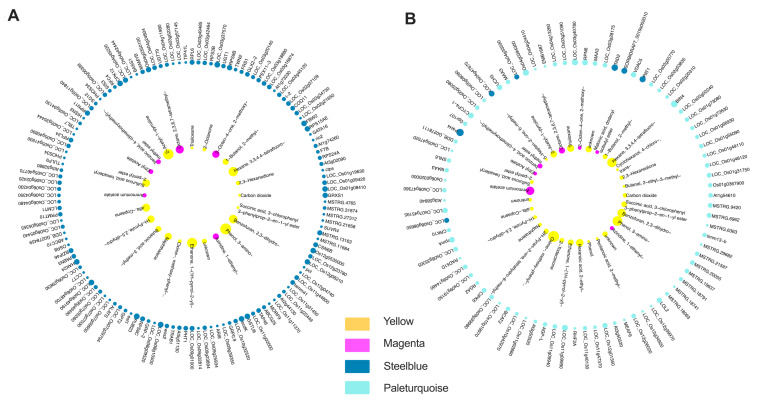
Correlation between transcripts and metabolites. (**A**) Significant positive correlations between selected transcripts and metabolites. (**B**) Significant negative correlations between selected transcripts and metabolites. Transcripts were from the steel blue and pale turquoise modules, while metabolites were from the yellow and magenta modules. Nodes are depicted as circles with corresponding colors, with the size of the circle proportional to the number of links. Red lines represent positive correlations, while green lines represent negative correlations.

**Table 1 ijms-25-08207-t001:** The significant different metabolites between SSXN and HKG rice varieties in the four stages.

Metabolites	Seedling Stage ^+^			Reproductive Stage			Filling Stage			Maturation Stage		
	Log2FC	*p* Value	VIP *	Log2FC	*p* Value	VIP	Log2FC	*p* Value	VIP	Log2FC	*p* Value	VIP
2-Acetyl-1-pyrroline	23.858	0.000	1.618	23.627	0.000	1.551	22.953	0.002	1.729	24.424	0.000	2.019
(+)-epi-Bicyclosesquiphellandrene	−2.133	0.007	1.243	−1.837	0.005	1.277	2.767	0.000	1.897	16.211	0.003	1.914
Ethanone, 1-(1H-pyrrol-2-yl)-	19.915	0.000	1.614	19.851	0.000	1.587	19.097	0.003	1.681	20.720	0.001	1.911
Phenol, 3-amino-	19.218	0.001	1.394	19.441	0.001	1.375	18.384	0.002	1.715	19.444	0.001	1.966

* VIP: The Variable Importance of Projection. ^+^ Stages: The stages include the four stages of rice development, which are seedling stage (S), reproductive stage (R), grain filling (F), and maturation (M).

**Table 2 ijms-25-08207-t002:** The fragrant metabolites in magenta module.

Metabolite ID	Description	Flavor	Class
Analyte84	2-Butanol, (R)-	alcoholic, sweet, fruity	alcohol
Analyte428	2-(3-Methyl-but-1-ynyl)-cyclohexene-1-carboxaldehyde	jasmine	aldehydes
Analyte90	Butanoic acid, ethyl ester	pineapple-like	ester
Analyte55	Propanoic acid, 2-methyl-, ethyl ester	fruity	ester
Analyte33	Ethyl Acetate	fruity, sweet, pear-like	ester
Analyte94	Carbonic acid, ethyl-, methyl ester	sweet, ether-like	ester
Analyte502	Benzoic acid, ethyl ester	sweet, fruity, cherry, grape	ester
Analyte587	Benzeneacetic acid, ethyl ester	sweet, floral, fruity	ester
Analyte285	Hepten-2-yl tiglate, 6-methyl-5-	fruity	ester
Analyte828	Linoleic acid ethyl ester	faintly fruity	ester
Analyte431	gamma-Nonalactone	coconut, creamy, fruity	ester
Analyte233	2-Octen-4-one, 2-methoxy-	sweet, mushroom-like, earthy	ketones
Analyte130	4-Methylheptane-3,5-dione	buttery, creamy	ketones

**Table 3 ijms-25-08207-t003:** The fragrant metabolites in yellow module.

Metabolite ID	Description	Flavor	Class
Analyte188	1-butanol, 3-methyl-	blue cheese	alcohol
Analyte187	1-butanol, 2-methyl-	blue cheese	alcohol
Analyte121	1-propanol, 2-methyl-	sweet, musty	alcohol
Analyte102	butanal, 2-ethyl-3-methyl-	cocoa	aldehydes
Analyte566	naphthalene, 1,2,4a,5,8,8a-hexahydro-4,7-dimethyl-1-(1-methylethyl)-, [1S-(1,4a,8a)]-	woody	aromatic compound
Analyte218	styrene	sweet odor	aromatic compound
Analyte387	pyrrole	nutty odor	aromatic compound
Analyte268	2-acetyl-1-pyrroline	popcorn and cracker-like	aromatic compound
Analyte375	.alfa.-copaene	balsamic, woody, sweet	aromatic compound
Analyte129	benzene, 1,3-dimethyl-	aromatic	aromatic compound
Analyte165	o-xylene	faint sweet	aromatic compound
Analyte34	furan, 3-methyl-	chocolate	aromatic compound
Analyte135	p-xylene	faint sweet	aromatic compound
Analyte250	hexanoic acid, 3-hexenyl ester, (Z)-	sweet, fatty, grassy	ester
Analyte635	2,2,4-trimethyl-1,3-pentanediol diisobutyrate	mild and sweet	ester
Analyte75	isobutyl acetate	sweet, banana-like	ester
Analyte132	1-butanol, 3-methyl-, acetate	sweet, banana-like	ester
Analyte166	2-heptanone	sweet, fruity, and camphor-like.	ketones
Analyte92	3-pentanone, 2,4-dimethyl-	sweet	ketones
Analyte57	2,3-butanedione	buttery	ketones
Analyte286	2-hydroxy-3-pentanone	Sweet, fruity	ketones
Analyte137	2,3-hexanedione	butter, caramel, creamy,	ketones
Analyte678	ethanone, 1-(1H-pyrrol-2-yl)-	popcorn-like	ketones
Analyte604	2-tridecanone	milky, coconut, nutty	ketones
Analyte101	2,3-pentanedione	butter	ketones
Analyte433	bicyclo[3.1.1]heptan-2-one, 6,6-dimethyl-, (1R)-	woody	ketones
Analyte273	2-hydroxy-3-hexanone	sweet, fruity	ketones
Analyte377	2-decanone	floral, fatty, and peach-like	ketones
Analyte269	5-hepten-2-one, 6-methyl-	fruity and apple-like	ketones
Analyte239	2-octanone	fruity, cheese-like	ketones
Analyte6	pentane, 3-methyl-	sweet odor	saturated hydrocarbons

## Data Availability

All data analyzed in this study are included in the main manuscript and its additional files. The data supporting the findings of this work are available in the paper and its Appendix A. The transcriptome data that support the findings of this study have been deposited to the National Center for Biotechnology Information (NCBI) Sequence Read Archive (SRA) with the accession code PRJNA1103873.

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
