# Peer review of "Metabolome and Transcriptome Unveil the Correlated Metabolites and Transcripts with 2-acetyl-1-pyrroline in Fragrant Rice"

_ijms, 2024, doi:10.3390/ijms25158207_

Round 1

Reviewer 1 Report

Comments and Suggestions for Authors

Comments and Suggestions for Authors

Dear Author,

I have an honor to review the manuscript entitled “Comparative metabolome and transcriptome analysis of fragrant rice unveils the correlated metabolites and transcripts 3 with 2-acetyl-1-pyrroline (2AP)” a research article submitted to MDPI Journal, International Journal of Molecular Sciences. Authors of this manuscript conducted comparative metabolome and transcriptome analyses between fragrant and control rice cultivars and identified strong fragrant rice cultivars ShangsiXiangnuo. They have extracted Flavonoid and performed several biochemical tests to detect differential metabolomes. Further, identified differentially expressed genes through transcriptome analysis. They predicted the results from the both analyses as well detected the mutation by sequencing. Overall, the experiments are performed well and the results are convincing. Thus, the presented results take up an important topic consistent with the profile of the Journal.

-However, even, manuscript is well organized and well described of the conception, I have some suggestions, which might improve the manuscript to make important to the wider reader.

-This article required firm aim of the study that should be underlined precisely and simultaneously and highlight why this is important to study.

-There are many places where grammar can be improved. I suggest a careful revision

-Few suggestions I have mentioned in the main text pdf file. Please check

Title:

Title is wordy and lengthy

Abstract: -Good organization with results order.

Introduction: Nicely presented

-However, need improvement with more informative specific findings referencing recent publications. 

2. Results

Results are well organized and nicely presented

3. Discussion

Consistent with results

4. Methods

I have some suggestions in the pdf

Comments on the Quality of English Language

Author Response

Dear Reviewer 1,

Thank you for your positive feedback on our manuscript. We appreciate your recognition of our work and the relevance of our findings to the journal's profile. We will address your suggestions to improve the manuscript further.

Comments 1: However, even, manuscript is well organized and well described of the conception, I have some suggestions, which might improve the manuscript to make important to the wider reader.

Responses 1: Thank you for your suggestions. We appreciate the suggestions of enhancing the manuscript which would make our manuscript more impactful for a wider audience. We have carefully considered your comments and made the necessary revisions.

Comments 2: This article required firm aim of the study that should be underlined precisely and simultaneously and highlight why this is important to study.

Responses 2: We have revised the introduction section to clearly state the aim of the study.

Comments 3: There are many places where grammar can be improved. I suggest a careful revision. Few suggestions I have mentioned in the main text pdf file. Please check.

Responses 3: Thank you for providing detailed feedback. We had revised the manuscript according to the PDF file. And we also thoroughly reviewed the manuscript for grammatical errors. Several corrections were made in the revised manuscript.

Comments 4: Title is wordy and lengthy

Responses 4: We change the title into “Metabolome and transcriptome unveil the correlated metabolites and transcripts with 2-acetyl-1-pyrroline in fragrant rice”.

Comments 5: Abstract: -Good organization with results order.

Responses 5: Thank you for your positive feedback.

Comments 6: Introduction: Nicely presented. However, need improvement with more informative specific findings referencing recent publications.

Responses 6: We have enhanced the introduction by including more specific findings and references to recent publications. This provides a stronger context for the study and highlights its relevance to current research.

Comments 7: Results are well organized and nicely presented

Responses 7: Thank you for your positive feedback on the results section.

Comments 8: Discussion: Consistent with results

Responses 8: Thank you for your positive feedback on the discussion section.

Comments 9: Methods: I have some suggestions in the pdf

Responses 9: We have reviewed the suggestions provided in the PDF and made the necessary revisions to the methods section. Your input has been very helpful in improving the clarity and completeness of this section.

Thank you once again for your valuable comments, which will contribute to the overall improvement of our study. Please let us know if there are any additional revisions or clarifications needed.

Reviewer 2 Report

Comments and Suggestions for Authors

The manuscript is well-written and clearly presented.

I have some comments are needed to be addressed before considering it for publication.

- Please double-check the data presented in Fig1B. It should be an 806 bp deletion (Line 110; Fig1B). The deletion of 679 bp in intron 4 should also be presented in Fig1B such as 86 bp deletion in exon 4 and 23 bp deletion in exon 5. Revise and present with re-corrected Fig1B. In addition, I see no data from Fig1A was cited/mentioned in the section “2.1. Rice Fragrance Confirmation through Sensory and Genetic Analysis”. The BADH2 gene in SSXN should also be directly embraced in Fig 1A.

- Provide with following information:

+ Explanation text for the abbreviation of “VIP” and put in the table footer.

+ The gene name for the supplementary table S8 and gene symbol of LOC_Os04g23140.

A few minor remarks:

- The full form of the abbreviation “WGCNA” = Weighted Gene Coexpression Network Analysis should be presented since it was firstly mentioned (Line 24).

- Line 99-100: “we employed WGCNA analysis to investigate the relationships within metabolites or genes” should be rewritten as “we employed WGCNA analysis to investigate the relationships between metabolites and transcripts/gene expression.”

- Lowercase these words since they are common nouns: Seedling, Reproductive, Fill, and Maturation (Line 18-19); Gas Chromatography; Mass Spectrometry; Volatile Aroma Compounds ((Line 77-79); Differentially Expressed Metabolites (Line 171), the 2nd paragraph in section 2.4 (Line 232-250); and elsewhere need double-check.

- Recorrect the section “2.3.2” to “2.3”. Italicize and Capitalize Each Word in its text (Line 153).

- Italicize the gene name: BADH2 (Line 109, 322, 326, 329), Badh2 (Line 433), and scientific name: “Oryza sativa” (Line 497)

- The legend of Fig3 should be not separated (Line 200-201)

- Present abbreviation as “SSXN” instead of its full form “ShangsiXiangnuo” (line 434, 446) since it was first mentioned on line 14

- Change: “milligrams” (Line 452) to “mg”; “microliters” (Line 453) to “µL”.

- Recorrect: mL min−1 to the superscript of “−1” (Line 462, 464)

- Provide the material/kit was employed to conduct cDNA synthesis which presented in line 486

- Unclear meaning of “this process” (Line 495)

- Section “4.6. Data analysis” should be put after the section “4.7. Quantitative real-time PCR validation”.

- Line 319: “Figure 1” change to “Figure 1B.

-  In the legend of figures, the figure panel should be listed as (A), (B) in bold format instead of A, B.

- Double-check the manuscript format follows the journal's instructions. The title and subsection (2.3, 2.6, and 4.1 to 4.7) should be Capitalize Each Word as this has been done for section 2.1.

For example: Section "2.6. The correlation between metabolome and transcriptome" -> "2.6. The Correlation Between Metabolome and Transcriptome "

Author Response

Dear Reviewer 2,

Thank you for your thorough assessment of our manuscript and for providing valuable feedback. It’s happy to receive your review report. We have addressed each of your comments below:

Comments 1: Please double-check the data presented in Fig1B. It should be an 806 bp deletion (Line 110; Fig1B). The deletion of 679 bp in intron 4 should also be presented in Fig1B such as 86 bp deletion in exon 4 and 23 bp deletion in exon 5. Revise and present with re-corrected Fig1B. In addition, I see no data from Fig1A was cited/mentioned in the section “2.1. Rice Fragrance Confirmation through Sensory and Genetic Analysis”. The BADH2 gene in SSXN should also be directly embraced in Fig 1A.

Responses 1: Thank you for your detail suggestion. We presented the whole 806 bp deletion with highlighting the exon 4 and the exon 5 in the revised Fig 1B. The citation of Fig 1A was also added in the 2.1 section.

Comments 2: Explanation text for the abbreviation of “VIP” and put in the table footer.

Responses 2: OK. The abbreviation “VIP” has been explained in the table footer as "Variable Importance in Projection."

Comments 3: The gene name for the supplementary table S8 and gene symbol of LOC_Os04g23140.

Responses 3: We updated the corresponding gene names in the supplementary table S8. The gene LOC_Os04g23140 is an expressed uncharacterized protein.

Minor remarks:

Comments 4: The full form of the abbreviation “WGCNA” = Weighted Gene Coexpression Network Analysis should be presented since it was firstly mentioned (Line 24).

Responses 4: The full form of the abbreviation “WGCNA” = Weighted Gene Coexpression Network Analysis has been presented the first time it is mentioned (Line 24) in the Abstract.

Comments 5: Line 99-100: “we employed WGCNA analysis to investigate the relationships within metabolites or genes” should be rewritten as “we employed WGCNA analysis to investigate the relationships between metabolites and transcripts/gene expression.”

Responses 5: Thank you for that suggestion. We had revised the manuscript according to this comment.

Comments 6: Lowercase these words since they are common nouns: Seedling, Reproductive, Fill, and Maturation (Line 18-19); Gas Chromatography; Mass Spectrometry; Volatile Aroma Compounds ((Line 77-79); Differentially Expressed Metabolites (Line 171), the 2nd paragraph in section 2.4 (Line 232-250); and elsewhere need double-check.

Responses 6: There has been changed into lowercase words. And we had double-checked throughout the manuscript for the misusing uppercase words.

Comments 7: Recorrect the section “2.3.2” to “2.3”. Italicize and Capitalize Each Word in its text (Line 153).

Responses 7: That had been revised as suggestion.

Comments 8: Italicize the gene name: BADH2 (Line 109, 322, 326, 329), Badh2 (Line 433), and scientific name: “Oryza sativa” (Line 497)

Responses 8: We had checked the whole manuscript to revise the names that should italic.

Comments 9: The legend of Fig3 should be not separated (Line 200-201)

Responses 9: OK, thank you.

Comments 10: Present abbreviation as “SSXN” instead of its full form “ShangsiXiangnuo” (line 434, 446) since it was first mentioned on line 14

Responses 10: OK.

Comments 11: Change: “milligrams” (Line 452) to “mg”; “microliters” (Line 453) to “µL”.

Responses 11: OK.

Comments 12: Recorrect: mL min−1 to the superscript of “−1” (Line 462, 464)

Responses 12: OK.

Comments 13: Provide the material/kit was employed to conduct cDNA synthesis which presented in line 486

Responses 13: The kit information had been inserted.

Comments 14: Unclear meaning of “this process” (Line 495)

Responses 14: “This process” means the filtration and rRNA sequence removal. We replaced “this process” into “filtration and rRNA sequence removal” in the sentence.

Comments 15: Section “4.6. Data analysis” should be put after the section “4.7. Quantitative real-time PCR validation”.

Responses 15: OK.

Comments 16: Line 319: “Figure 1” change to “Figure 1B.

Responses 16: OK.

Comments 16: In the legend of figures, the figure panel should be listed as (A), (B) in bold format instead of A, B.

Responses 16: OK. In the legends of figures, the figure panels are now listed as (A), (B) in bold format.

Comments 17: Double-check the manuscript format follows the journal's instructions. The title and subsection (2.3, 2.6, and 4.1 to 4.7) should be Capitalize Each Word as this has been done for section 2.1.

Responses 17: OK. Thank you.

Thank you once again for your valuable comments, which will contribute to the overall improvement of our study.